



# Consistent variability but different spatial patterns between observed and reanalysed sea-ice thickness

Joula Siponen[1,2], Petteri Uotila[1], Eero Rinne[2], and Steffen Tietsche[3]

[1]Institute for Atmospheric and Earth System Research, University of Helsinki
[2]Finnish Meteorological Institute
[3]European Centre for Medium-Range Weather Forecasts

**Correspondence:** Petteri Uotila (petteri.uotila@helsinki.fi)

**Abstract.**

Changes in sea-ice thickness are one of the most visible signs of climate change. However, to gain a comprehensive understanding of mechanisms involved, long time series are needed. Importantly, the development of more accurate predictions of sea ice in the Arctic requires good observational products. To assist this, a new sea-ice thickness product by ESA Climate
Change Initiative (CCI) is here compared to the ocean reanalysis ORAS5 by ECMWF for the first time. The CCI product is based on two satellite altimetry missions, CryoSat-2 and ENVISAT, which are combined to the longest continuous satellite altimetry time series of Arctic-wide sea-ice thickness, 2002–2017 and continuing.

Time series of sea-ice volume for the CCI coverage reveal years of extremely low volume as well as recovery during the winter season. The 15-year trends in sea-ice volume are clearly negative over the time series and despite large variability
between years statistically significant. The 15-year ORAS5 trends have larger interannual variability than the CCI trends and are therefore not statistically significant despite of a good match in terms of year-to-year variability. The observed negative trends result from changes in both atmospheric and oceanic forcing.

The CCI product performs well in the validation of the ORAS5 reanalysis: overall root-mean-square difference (RMSD) between sea-ice thickness from CCI and ORAS5 is below 1 m. However, seasonal and interannual RMSD variations during the
time series are large, from 0.5 m to 1.3 m. The differences are a sum of reanalysis biases, such as incorrect physics or forcing, as well as uncertainties in satellite altimetry, such as the snow climatology used in the thickness retrieval.

## 1 Introduction

Climate change can be seen in the Arctic more clearly than anywhere else in the world. Mean near-surface air temperature has
increased by 2.5 °C since the end of the 20th century, which is double the global average rate (Overland et al., 2017). Related to this, sea-ice cover in the Arctic is experiencing major changes, which reflect back to the global climate system (e.g. Overland et al., 2017; Meier, 2017; Serreze and Meier, 2018).



Multiple studies have shown that climate models tend to underestimate the rate of sea-ice loss compared to observations (e.g. Stroeve et al., 2007). The latest IPCC report in 2013 showed that still only one fourth of the CMIP5 climate models reproduce the trend in September sea-ice extent as strong or stronger than observations (Flato et al., 2013). Ding et al. (2018) analysed a set of fully coupled climate models together with sea-ice concentration observations, and concluded that the models lack in simulating the linkages between the Arctic and lower latitudes.

Sea ice in the Arctic is thinning and getting younger as the multiyear ice is melting (Maslanik et al., 2007). The change was noticed already in the 1990s (Rothrock et al., 1999) but since then the sea-ice decline has accelerated and the totally ice free Arctic Ocean in summer is getting very close (Serreze and Meier, 2018). Laxon et al. (2003) showed that numerical models and volume measurements agree that the sea-ice thickness varies on much longer timescales than ice extent, which makes it a better indicator of climate change. Changes in sea-ice volume also tell about the changes in the Arctic heat budget as well as freshwater exchange between sea ice and the ocean.

In this study, a new product of sea-ice thickness based on radar altimetry is used. The product covers time series of 2002–2017 and became available in summer 2018 as a result of the ESA (European Space Agency) Climate Change Initiative. The processing methods used for the freeboard retrieval from ENVISAT was developed by Paul et al. (2018). Previously, Guerreiro et al. (2017) and Schwegmann et al. (2015) have done analysis of the overlapping period between Cryosat-2 and ENVISAT both in the Arctic and Antarctic showing that the overlapping period can help to improve the sea-ice thickness data set by ENVISAT.

The new satellite product is compared to the ORAS5 reanalysis by the European Centre for Medium-range Weather Forecasting (ECMWF), which provides the initial conditions for all ECMWF re-forecasts (Johnson et al., 2019; Zuo et al., 2019). Zuo et al. (2018) concluded that the amount of assimilated data correlates with the quality of the reanalysis output – the more assimilated data, the better the result. A lack of observational products with proper quality control and coverage has prevented the assimilation of sea-ice thickness to the reanalysis, but it has been suggested that it could increase the skill of forecasts significantly (Day et al., 2014).

Previously, ORAS5 reanalysis and its pilot version ORAP5 have been validated with multiple observational sea-ice thickness data sets, including ICESat and SMOS observations (e.g. Tietsche et al., 2017b, a; Zuo et al., 2019), but this is the first comparison with the ESA CCI sea-ice thickness data set combining CryoSat-2 and ENVISAT missions. This new data set gives unique insight into the development of sea-ice thickness in the Arctic over a time series of 15 years from 2002 to 2017, during which the Arctic has gone through interesting years climatologically.

The aim of the study is to give an answer to the question: Can the ESA CCI sea-ice thickness product be used for the validation of sea ice in the ORAS5 ocean reanalysis during the growth season? To answer this question, the mean sea-ice thickness as well as trends in sea-ice thickness and sea-ice volume are compared, and their uncertainties are taken into account. In addition, regional differences in sea-ice thickness are examined and linked to known challenges in modelling of sea ice as well as in satellite altimetry.



## 2 Data sets of sea-ice thickness

### 2.1 ESA CCI sea-ice thickness

The satellite radar altimetry dataset for sea-ice thickness used here is developed by Alfred-Wegener-Institut (AWI) Helmholtz Zentrum für Polar und Meeresforschung and Finnish Meteorological Institute as part of the ESA Sea ice Climate Change Initiative (CCI) project Phase 2. The CCI sea ice thickness processing chain from Level 1b altimeter data and auxilliary data sets is described in detail in Paul et al. (2017). The data set provides Level 3 (L3) sea-ice thickness measurements from the Northern Hemisphere winter months (October–April) 2002–2017 in monthly averaged grids (Table 1).

Auxiliary datasets used in the processing of the satellite altimetry measurements into sea-ice thickness are a snow depth climatology by Warren et al. (1999), the DTU15 global mean sea surface (Andersen and Knudsen, 2009), OSI-SAF Global Sea-Ice Concentration (EUMETSAT, 2015) and sea-ice type from ESA-SICCI multi-year area fraction based on National Snow and Ice Data Center (NSIDC) SSMI-SSMIS brightness temperature (Version 4).

An updated and comprehensive description of estimating sea-ice thickness and volume with satellite altimetry (CryoSat-2) is provided by Tilling et al. (2017). The original CryoSat mission and its design and operation have been described by Wingham et al. (2006). Before the CryoSat mission, other altimetry satellites were in use. One of them was ENVISAT satellite carrying RA-2 (Radar Altimeter) operating from 2002 to 2012. CryoSat-2 measures sea ice mostly in SAR (Synthetic Aperture Radar) mode utilising delay Doppler processing, which leads to major improvements over ENVISAT – most importantly the smaller along-track footprint of only 300 m. CryoSat-2 has been shown to agree well with in-situ observations and provides a realistic seasonal cycle of the growing season of sea ice in the Arctic (Laxon et al., 2013). ENVISAT has suffered from large biases related to the mentioned footprint characteristics, but the overlapping period with CryoSat-2 has enabled improvements of the data set based on their differences as well as advanced retracking algorithms (Paul et al., 2018).

The conversion of sea-ice freeboard to sea-ice thickness requires knowledge of the densities of sea ice, sea water and snow, as well as the snow thickness on ice. These are sources of uncertainty not related to the radar signals and their processing. The difference in the performance of the snow climatology by Warren et al. (1999) over first year ice (FYI) and multi year ice (MYI) has lead to the snow depth to be corrected by taken only 50% over FYI (Laxon et al., 2013). This approach is far from solid and requires additional sea-ice type data. Effort has been recently put to improving snow depth estimates. Blanchard-Wrigglesworth et al. (2018) reconstructed the snow in the Arctic using observations of sea-ice movement as well as snow precipitation from reanalysis. Their results show high interannual variability, so snow on sea ice is certainly one of the most important sources of uncertainty in sea-ice thickness estimates from altimetry, also for climatological estimates.

### 2.2 ORAS5 Ocean reanalysis

ORAS5 is a global eddy-permitting ocean-sea-ice reanalysis that includes estimates of the ocean and sea-ice states from 1979, when continuous satellite passive microwave measurements of sea-ice concentration began, to near-real-time (NRT). Together with the real-time analysis component it forms the OCEAN5 system that is used to initialize the state of the ocean and sea ice





in ECMWF's operational predictions of different time scales. See Zuo et al. (2019) for a full documentation of the OCEAN5 system and validation results.

ORAS5 comprises the NEMO (Nucleus for European Modelling of the Ocean) ocean model (Madec et al., 2016) and the LIM2 (Louvain-la-Neuve Sea-Ice Model, version 2) sea-ice model (Fichefet and Maqueda, 1997), which is coupled to the ocean model every three time steps. The grid of the model is tripolar, and the resolution of the model configuration is 0.25 degree at the equator. In the Arctic the resolution can be better than 5 km in some areas, for example in the Canadian Arctic Archipelago due to the location of the three poles of the grid.

Atmospheric forcing of ORAS5 comes from ERA-Interim (Dee et al., 2011) before 2015 and from operational ECMWF analysis after that. In addition, ORAS5 assimilates the following observations: sea-surface temperature from HadISST2.1 before 2008 (Titchner and Rayner, 2014) and OSTIA (Donlon et al., 2012b)) afterwards, sea-ice concentration from OSTIA, in-situ temperature–salinity profiles from EN4 (Gouretski and Reseghetti, 2010), and sea-level anomalies from AVISO (Pujol et al., 2016). The data assimilation system used is NEMOVAR (Mogensen et al., 2012) in a 3D-Var-FGAT configuration.

Sea-ice thickness is not assimilated to ORAS5.

The original model output consists of 5 ensemble members with slightly different initial conditions, surface forcing and assimilated observations. Due to stratified random sampling method the members see different observations. However, for sea-ice variables the spread between the ensemble members is small (Zuo et al., 2019), so we consider only the central ensemble member here. For temporal and spatial comparability, the daily data were averaged to monthly means and regridded to the CCI

satellite data grid EASE that is a Lambert Azimuthal grid with $25 \times 25$ km equal cell size. The regridding was done using the Climate Data Operators (CDO) tool (info and documentation at https://code.mpimet.mpg.de/projects/cdo).

## 3   Comparison methods

### 3.1   RMSE and correlation

The validation of ORAS5 sea-ice thickness dataset with respect to ESA CCI sea-ice thickness product was conducted including

both temporal and spatial comparison. For this purpose two methods were used: (1) Root Mean-Square Error (RMSE), which is a measure telling how well predicted values by model match with observations, and (2) correlation of the mean sea-ice thickness for autumn and spring seasons separately. Additionally, the difference maps for specific months, as well as the average difference in autumn and spring season, were compared. The formulation for RMSE is the following:

$$RMSE = \sqrt{\frac{\sum_{i=1}^{N} (z_{r_i} - z_{o_i})^2}{N}} \tag{1}$$

where $N$ is the number of grid cells, $z_{r_i}$ the ocean reanalysis grid of sea-ice thickness and $z_{o_i}$ the observations. The advantage of RMSE is that it gives one value for each time step representing the mean squared difference between datasets in hand. However, it does not tell the sign of the difference, neither which one of the sets represents the reality better. Tietsche et al.





(2014) studied RMSE of ORAP5 Ocean Reanalysis (the pilot version of ORAS5) with respect to ICESat (laser altimetry satellite by NASA) sea-ice thickness. In their work ICESat data from 15 missions between 2003 and 2008 were interpolated to
the model grid, and only grid cells with ICESat thickness larger than 0.5 m were considered. In this work the model data were interpolated to the satellite observation grid.

The second method is looking at the correlation. The mean thickness grid of ORAS5 and CCI over the Arctic were calculated for each month separately, and then the correlation between grid cells that had valid CCI data. $R^2$ was used to measure the correlation, R being the correlation coefficient from linear regression. In order to compare correlation in different areas, the
Arctic was divided into 10 regions simply by longitude (see Table 2). In the north the limit is 81.5 °N, because of the satellite track of ENVISAT.

### 3.2 Sea-ice volume

Since the data from satellite measurements are restricted to area south of approximately 81.5 °N, the central Arctic Ocean is masked out from the ORAS5 data and sea-ice volume is calculated for the area between 45 °N and 81.5 °N. Sea-ice thickness
data from satellite observations only include grid cells with sea-ice concentration more than approximately 75 %. Smaller sea-ice concentration leads to errors in freeboard measurements because the radar echoes from sea ice and open water form similarly diffuse power spectrum. From the ORAS5 data the cells that had coverage in the CCI dataset were taken into account to have a comparable time series.

$$SIV = \sum_{i=1}^{N} SIT_i \times SIC_i \times A \tag{2}$$

Additional data are needed in order to calculate the sea-ice volume by multiplying sea-ice thickness, $SIT$, with area, $A$, and sea-ice concentration, $SIC$, of the grid cell $i$ as the equation 2 shows. The ORAS5 data were converted to the same grid with the CCI data, that have constant grid size of $25 \times 25$ km. This makes comparison straightforward. Sea-ice concentration again is included in both data sets, and it comes originally from the same observational passive microwave data set (OSI-SAF/OSTIA (Donlon et al., 2012a)). Same data source for SIC leads to results being less independent.

In the light of the declining trend in sea ice in the Arctic, the volume changes of Arctic sea ice were studied and the ability of the model to reproduce the interannual variability of sea-ice volume as well as possible trends for the time series of 10/2002–4/2017 were looked at. The analysis was carried out separately for each month so that each time series is continuous and the differences between phases of the growing season can be studied. The monthly trends were calculated by linear least-squares regression and the statistical significance of the trends based on p-value that uses null hypothesis of no trend. The trends were
included to the comparison, marked differently if insignificant.





## 4 Results

### 4.1 RMSE and the difference geographically

Root-Mean-Square-Error (RMSE) calculated for ORAS5 and CCI sea-ice thickness as a time series from 2002 to 2017 monthly is shown in Figure 1. It includes the CCI data coverage. The mean RMSE over the ENVISAT period, 2002–2012, was 0.88

m. Over the CryoSat-2 period, 2010–2017, it was 0.73 m for the coverage up to 88 °N and 0.78 m when limited to 81.5 °N. The maximum RMSE over the whole time series was almost 1.3 m in October 2014 ($< 81.5$ °N) and the minimum close to 0.5 m in the beginning of CryoSat-2 time series in November 2011. Time series of CryoSat-2 starting from 2010 follows the same seasonal curve in RMSE as the overlapping ENVISAT period 2010–2012 but the values are approximately 0.1 m smaller. Taking the central Arctic into the RMSE calculation for CryoSat-2 time series does not improve the results substantially except

for some specific months, such as extremely high RMSE in October 2014.

There is a clear seasonal variation recurring yearly that forms an u-shape. RMSE is the smallest usually (12 out of 15 years) in November or December, and largest in February or March (10 out of 15). Some years (2 of 15) the highest RMSE is in October. RMSE increases from November to March almost every year. Specific months of the time series are studied with sea-ice thickness difference maps in order to investigate the spatial variability of the difference between ORAS5 and CCI sea-ice

thickness. The average, the worst and the best autumn (Figure 2) and spring (Figure 3) maps are shown. The best and the worst, based on RMSE, are selected from the Cryosat-2 time period so that they include also the central Arctic.

Figure 2 shows the difference between ORAS5 and altimetry measurements in autumn. The maps reveal the yearly difference in how well the reanalysis agrees with the satellite measurements in the beginning of the growing season as well as the average difference. The amount of valid satellite altimetry data points is smaller compared to spring season. November 2014 has one of

the largest RMSE values, ca. 0.85 m, which means that the data sets disagree in average almost one meter. As the map shows, there are persistent differences of both negative and positive sign depending on the region. November 2016 shows one of the lowest RMSE values, 0.55 m. Regional features of the differences are similar in both of the cases: north of Greenland and in the Beaufort Sea the ice is clearly thicker in ORAS5 than CCI satellite observations show, and there are patches of thicker ice by CCI altimetry observations along the eastern coast of Greenland, in the central Arctic as well in the Baffin Bay. The patterns

are strongly present in the average map too.

In 2014 the thickness difference between the data sets is up to 2 meters from north of Greenland to Beaufort Sea. The sign of the difference can not be seen from RMSE time series but the geographical maps reveal more. In some areas the CCI product shows up to 2 meters thicker ice compared to ORAS5. For example north of Canadian Arctic Archipelago in 2016, unlike 2014, there is an area where the reanalysis is underestimating the thickness compared to observations.

Figure 3 illustrates the difference in the end of the season. March has the largest number of valid satellite data points during the growth season every year. Maps show two interesting features. First, ORAS5 seems to overestimate the thickness in the Beaufort Sea even more than in the beginning of the season, at least 2 meters, but the bias has moved against the Alaskan coast and towards Siberian side. From north of Svalbard to the central Arctic there is an equally strong negative bias visible in both





years and in average. Second, the Bering Sea, Hudson Bay and Baffin Bay regions show slightly positive bias as well, unlike
in the beginning of the season.

### 4.2 Correlation of mean sea-ice thickness by month

Correlation between the mean sea-ice thickness over the years 2002–2017 by ORAS5 and CCI product is shown in the Figures
4 and 5. The American and Siberian side of the Arctic Ocean are both divided into five longitudinal sectors for which autumn
(Nov/Dec) and spring (Feb/Mar) thickness scatter plots are presented. Correlations vary between the seasons and the regions
185  so that the R-squared values range between 0.03 in spring in the Laptev Sea sector and 0.91 in autumn in the Norwegian Sea
section.

In the Siberian Arctic the correlation is decreasing as the growing season advances. The thickest ice of between 2 and
3 meters can be found in spring in the East Siberian, Laptev and Kara Seas. The sea ice in the Laptev and Kara Seas is
persistently thicker in ORAS5 than in CCI. Especially at the very coast of Siberia in the Laptev Sea sector there is a high
positive bias in the model compared to the observed thickness that increases during the growing season. Here the satellite
observations give very small thickness, close to no ice at all, when the model expects up to 3 meters thick ice. This is most
likely due to the model poorly simulating the opening of coastal polynyas. This same pattern was seen in the maps of spring
difference in sea-ice thickness (Figure 3). However, there are clear regional differences. In the Barents Sea, as well as the East
Siberian Sea, the slope of the fit is rather good in the autumn, despite a large amount of outliers. In the Norwegian and the
Barents Sea, there is thicker ice by the observations than the reanalysis, which differs from the other areas.

The sea ice on the American side of the Arctic is thicker overall. In the Chukchi and Bering Seas, the correlation is fairly
good in both seasons even though ORAS5 tends to have somewhat higher ice thickness than CCI, especially in spring (Figure
5). The Beaufort Sea and the Canadian Arctic Archipelago have the strongest positive bias by ORAS5, with similar magnitudes
as the bias in the Laptev Sea. The thickness in the Canadian Arctic Archipelago and the Hudson Bay is scattered with values
up to 5 m by ORAS5 but to 4 m by CCI. Similar to the Barents Sea in the Siberian side, the Greenland Sea is a sector with
notably thicker ice by the CCI product, up to 5 m in the spring, compared to the ORAS5 thickness up to approximately only 3
m. The situation in the Baffin Bay appears to be similar.

### 4.3 Sea-ice volume time series

To look at the time series of sea-ice conditions over the whole Arctic, the sea-ice volume (SIV) for the satellite data coverage,
between the latitudes 45 and 81 °N, was calculated following equation 2 for each month separately. As Figure 6 shows, SIV
varies in the beginning of the growing season (October) from 1500 to almost 4000 $km^3$ by CCI when by ORAS5 the upper
limit is closer to 5000 $km^3$. In the end of the season (March), the volume is between 11000 and 14500 $km^3$ by CCI and ORAS5
volume variation is roughly inside the same range. The variation is relatively smaller in the end of the season when there is
more ice.
Interannual variation and extreme years in the sea-ice volume are captured by both the reanalysis and the observations.
During the ENVISAT period the sea-ice volume decreases rather steadily, with the exception of a deep drop in 2007/2008




together with a subsequent recovery. The decreasing trend turns into peaking sea-ice volume during the seasons 2013/2014 and 2014/2015, during the CryoSat-2 time series. This peak is notably larger with ORAS5 than CCI, which can be seen also as the large RMSE values 2014/2015. In the last two seasons of the time series from 2015 to 2017, the volume drops close to
the level before the peak from October to February, and even below it in March and April.

Linear trends of SIV are clearly negative over the period 2002–2017 for both the CCI product (including CryoSat-2 and ENVISAT) and ORAS5. CCI trends are statistically significant for all the months, unlike ORAS5 trends. On its own, the CryoSat-2 trends are not statistically significant due to the short time series (up to 7 years) and anomalously high volumes compared to the whole time series. A large increase in volume in 2014–2015 is most probably behind insignificant ORAS5
trends.

CCI has the strongest negative trend in March and April, around -1200 km$^3$/decade, and November and December trends are not far behind (Table 3). The difference between CCI and ORAS5 trend is smallest in the beginning of the season and largest in February and March (up to 750 km$^3$/decade). In October, the reanalysis shows a more negative trend than observations, whereas in other months its trends are less negative than observed.

There is a peak in ORAS5 volume in spring 2012 (January–April), which is not visible in the CCI data. However, the major drop in the following year is captured by both ORAS5 and CCI even though their absolute volumes differ by up to 1000 km$^3$ in February and April. The largest continuous difference between CCI and ORAS5 is in April.

## 5  Discussion

### 5.1  Agreement with previous validation

Tietsche et al. (2014) compared ORAP5, the predecessor of ORAS5, with ICESat satellite laser altimetry sea-ice thickness and found mean RMSE of 0.93 m for the whole Arctic, which is up to 20 cm higher than the RMSE between CCI sea-ice thickness and ORAS5 over the CCI data coverage. The method of calculating the RMSE differs slightly, since Tietsche et al. (2014) interpolated the data from the ICESat missions over the whole Arctic before comparing all the data points that had sea-ice thickness of more than 0.5 m in the ICESat interpolation. We used only the grid cells that have CCI data coverage. This
leads to fewer and seasonally varying data points for comparison but requires no further interpolation on top of the averaging included in the processing of the CCI product. The advantage of CCI compared to ICESat is that the satellite radar altimetry produces data for all months of the growing season. ICESat has missions comprising 35 days in spring (February/March) and another 35 in autumn (October/November) (Kwok et al., 2007). Thus, the CCI product is a better validation product in the sense of temporal coverage.

Better resolution and coverage for the old ENVISAT mission have been achieved by developing the freeboard retrieval using the overlapping period of ENVISAT and CryoSat-2 satellite missions (Paul et al., 2018). Nevertheless, also the new product has to be used with caution. Paul et al. (2018) did a comparison of the freeboard difference of the ENVISAT and CryoSat-2 and they found still local differences up to 20 cm during the calibration period, which would be approximately 2 m in thickness by





assuming the hydrostatic equilibrium and excluding uncertainties related to snow. This difference signifies a moderately good
performance although individual pixels can have significant differences.

Coverage difference and better original resolution as well as more advanced instrumentation lead to better RMSE between
ORAS5 and CryoSat-2 than between ORAS5 and ENVISAT. The difference maps from three autumns (Figure 2) suggest that
in the beginning of the growing season the agreement is generally good in the Central Arctic, where February and March 2012
maps (Figure 3) reveal the much thicker ice in the CryoSat-2 measurements compared to ORAS5. Due to the inclination of the
satellite orbit these features cannot be studied from the ENVISAT data and they affect the RMSE. This can be seen as a slight
difference in RMSE between the total CryoSat-2 coverage and CryoSat-2 limited to 81.5 °N (Figure 1).

Comparisons between different reanalyses during the ENVISAT time period show very similar reanalysis biases than com-
parison done here. Results from Chevallier et al. (2017) for the mean thickness difference 2003–2007 between ORAP5 and
ICESat show the positive bias in the Beaufort Sea. Uotila et al. (2019) calculated the difference for mean thickness 2000–2012
between ORAP5 and ITRP thickness product that includes varying set of observations from airborne and satellite measure-
ments to upward looking sonars and submarines (Lindsay and Schweiger, 2015). Their maps showed also the Beaufort Sea
bias but the negative bias north of Svalbard as well. These are clearly visible from the results of this study, both from difference
maps and scatter plots (Figures 2, 3 and 5). The magnitude of these biases is the same, around 1.5–2 m towards the reanalysis
in the Beaufort Sea and towards the satellite observations north of Svalbard and along the ice edge in the Barents Sea.

## 5.2    Potential sources of biases in reanalysis and observations

Unrealistic performance of sea-ice thickness production in the model is a sign of inaccurate dynamics or thermodynamical
processes behind the sea-ice growth (Uotila et al., 2019). Other sources of erroneous results can be found in the atmospheric
and oceanic forcing, which often dominates the sea-ice evolution, as well as data assimilation. In any case, the chaotic nature
of the climate system makes it simply impossible to model reality in a perfect manner (Notz and Bitz, 2017).
Sea-ice concentration from ORAS5 has been compared to the ESA CCI sea-ice concentration product, and the results
confirm the large biases in the East Greenland and Labrador Seas (Zuo et al., 2019). A source of these biases is said to be
related to the model errors in the sea-ice model NEMO-LIM2 (Tietsche et al., 2014). The sea-ice thickness product of CCI
used here does not extent to the latitudes of the Labrador Sea (south of Greenland) but it shows the same bias in the East
Greenland Sea. That region is the only one considered here which has notably thicker sea ice in the satellite measurements
than in the reanalysis (see Figure 5). This is connected to the lack of thickness categories as well as inaccurate sea-ice drift in
the sea-ice model LIM2. Having only one thickness category and open water in the model, requires unphysical assumptions.
The realistic simulation of sea-ice concentration requires a multi-category thickness distribution inside a grid cell instead of
conductive heat fluxes calculated for an average thickness in the grid cell as in LIM2. The multi-category parameterisation
leads to more accurate simulation of sea-ice thickness below 0.5 m in the model. That limit is the initial thickness of sea ice
that forms in an open-water part of a grid cell in ORAS5, and is used for tuning of the model (Hibler, 1979). This could become
a bigger problem when the sea ice is getting thinner in most parts of the Arctic (Tietsche et al., 2014).





This effect of the thin ice error, in comparison of ORAS5 and CCI, is the most likely reason behind large RMSE values in the beginning of the growing season in October, as well as the thicker ice in the Barents Sea by ORAS5 in the spring. It can also explain part of the 3 m thicker ice in the Laptev Sea close to the Siberian coast. However, another major contributor to the
positive bias in the Laptev Sea can be appearance of land-fast ice and coastal polynyas that cannot be simulated by the model. Most of the ice along the Siberian coasts is often attached to the shore preventing movement and pressure ridge formation, which means that the ice cover is most likely thinner than in regions with moving sea-ice. In addition, polynyas often form at the edge of the land-fast ice when winds blow off-shore. These re-freeze quickly, resulting in large areas of thin ice (Tietsche et al., 2018). The satellite observations can be expected to perform moderately well in the region of less variable thickness
distribution, provided that there are some leads available for the reference sea surface height measurements.

The strong positive thickness bias in the Beaufort Sea occurs yearly throughout the time series. As mentioned earlier, this bias has been noticed to be a problem with this specific model but also others (e.g. Tietsche et al., 2014). The Beaufort Sea is containing many upward looking sonars used for validation of CCI data set meaning that especially in that area satellite altimetry can be assumed to perform well. According to Zuo et al. (2018), the model in this region has some biases in ocean
current and/or sea-ice velocities, which could partly explain why sea ice grows up to 2 m thicker in Beaufort Sea than CCI thickness suggests. The ice strength parameter is also lower compared to other reanalysis, which could produce too large drift speeds (Chevallier et al., 2017). The other, maybe even more profound, source of error can be related to the atmospheric forcing and unrealistically persistent high pressure in the area creating strong winds that push ice towards the Gyre. Serreze and Meier (2018) pointed out that the shift in atmospheric pressure patterns do not need to be large to affect the sea ice.

Model biases are one side of the story, but also satellite altimetry contains a lot of approximations and assumptions that accumulate uncertainty to the final thickness estimation, such as the constant density values for thickness retrieval. One problem is the instrument footprint, which can contain many different surfaces that all contribute to the received waveform. This means that the measurements are not always trustworthy close to land or inside archipelagos due to small amount of valid data point for monthly averaging. For example, in the Canadian Arctic Archipelago the amount of land possibly affects the measurements
or at least the amount of valid data to average over the month. This issue is more profound with ENVISAT mission that with CryoSat-2 due to the large footprint. On the other hand, large solid ice cover areas like land-fast zones with no leads for reference, can suffer from larger uncertainty.

Still, the largest acknowledged uncertainty in the satellite observations comes from the outdated snow product used for the conversion from freeboard to thickness. This is most likely behind the change in the sign of the difference between the model
and observations between February and March in the Hudson Bay area. This area lies far from the central and eastern Arctic where the snow climatology is based on. This is also the reason behind no data available from the Labrador Sea: Warren climatology gives even negative values there. Largely thicker observational values in the Greenland Sea are likely also related to the snow product. Thus, the difference of up to 5 m between CCI and ORAS5 can be attributed to a combination of too thick is by the satellite observations and too thin ice by the reanalysis.

Before a better snow product is found and included in the conversion of sea-ice freeboard to thickness (which should happen soon), the areas far from the central and eastern Arctic, such as the Hudson Bay, are to be taken with a grain of salt.





### 5.3 Interpretation of the volume time series

In this section we analyse the sea-ice volume time series from ORAS5 and CCI. The difference between the absolute volume values comes from sea-ice thickness as well as sea-ice concentration (that comes originally from the same source). Concen-
tration leads to some false correlation between the data sets that needs to be taken into account. Comparing the volume to any other sea-ice volume estimate over the Arctic would require interpolation of the CCI dataset to include also the very central Arctic Ocean as well as other missing data points. This has been done for the CryoSat-2 before and the results show similar annual volume development than PIOMAS model (Laxon et al., 2013).

As an improvement of the trend comparison with monthly gridded dataset, specific areas with good sea-ice coverage through-
out the season and time series could be chosen and volume time series of those areas calculated. The time series could be also detrended to confirm that the model can provide the anomalous events regardless of varying absolute values.

The peak in ORAS5 volume in the spring 2012 is not visible in CryoSat-2 or ENVISAT data to the same extent if at all, which leads to the question if the model would have been able to predict the record low sea-ice cover during the following summer 2012. Also the sea-ice volume of the season 2014–2015 is notably higher by ORAS5 than by the satellite product. The
large positive thickness bias along the Greenland and Canada/Alaska coast (Figure 3) persists from winter to spring (February–April, Figure 6) and due to thick snow on thick ice the ORAS5 melt and retreat are too slow at low Arctic latitudes. Why this bias is initially stronger in 2012 than in other years, could be due to atmospheric forcing errors: winds, temperature and large-scale flow.

Seasonal forecasts with SEAS5 substantially over-predict the summer minimum sea-ice extent when initialized from ORAS5
in spring. This has not been a major issue with the predecessor of the reanalysis, ORAS4, which includes a sea-ice climatology instead of thermodynamical model. (Johnson et al., 2019) The reason for the model not being able to melt the ice realistically can be explained partly by the ice being too thick in the beginning of the melting season. The comparison to CCI sea-ice thickness strongly supports this hypothesis. Observations covering the whole Arctic monthly for more than 15 years are extremely useful tool for validation and future development of ORAS5 but also other models and reanalysis products.

## 6 Conclusions

In this study the first comparison between ORAS5 ocean reanalysis and CCI sea-ice thickness data set has been conducted. Regardless of known issues and uncertainties, such as the old snow climatology, the CCI product performs well as a validation tool. It is better than ICESat due to its temporal and spatial coverage over the whole growing season.

The analysis of the RMSE together with difference maps and correlation of the mean thickness between the ORAS5 and
CCI reveal seasonal and regional differences that are more than 2 m in thickness, for example in the Beaufort Sea. Bias is not clearly towards thicker ice by the model but in some regions, such as the East Greenland Sea, the altimeter observes much thicker ice. Differences have a clear seasonal pattern that is connected to the model as well as forcing issues, and are consisted with earlier results. The development and validation of ORAS5 are important for the sake of improved seasonal forecasts.



The 15-year time series enables the calculation of long-term sea-ice volume trends. ORAS5 and CCI data set agree with each other as well as earlier results that the trends of all months of the growing season are negative. Moreover, they are in a good agreement in terms of year-to-year variability. However, the strong increase in volume 2013–2015 is much larger by ORAS5 reanalysis, which seems to be a new finding and reasons for it require further analysis.

*Data availability.* The ESA CCI ice thickness data are publicly available at http://cci.esa.int/content/cci-sea-ice-dataset-release-sea-ice-thickness-v20/, and the ORAS5 output from the Copernicus Marine Environment Monitoring Service under http://marine.copernicus.eu/.

*Author contributions.* ER, JS and PU contributed to the study design, while JS carried the analysis and wrote the paper. ER provided and assisted in the interpretation of the ESA CCI data and ST the ECWMWF ORAS5 output. All authors participated in paper preparation.

*Competing interests.* The authors declare that they have no conflict of interest.

*Acknowledgements.* The work of JS was financially supported by the Institute for Atmospheric and Earth System Research, University of Helsinki, and by the Finnish Meteorological Institute. Work of ER was partially funded by the ESA CCI+ sea ice project (contract number 4000126449/19/I-NB).





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





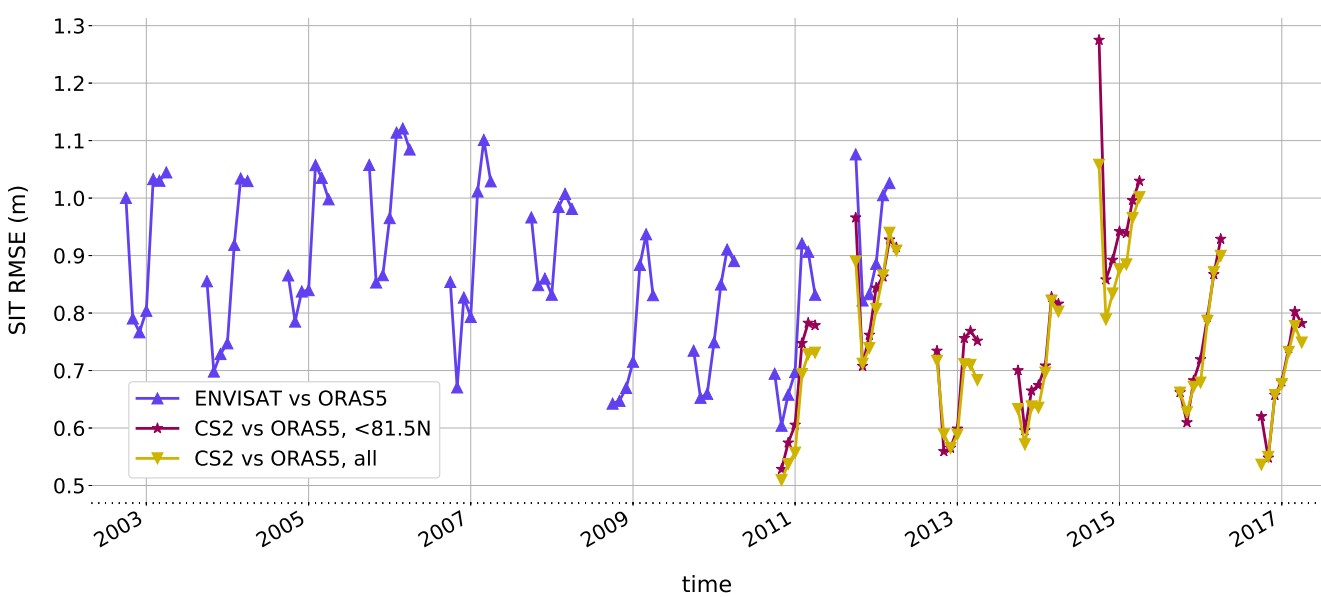

**Figure 1.** Root-Mean-Square-Error (RMSE) between ORAS5 and CCI October–April sea-ice thickness in the Arctic: between ENVISAT and ORAS5 in purple and CryoSat-2 and ORAS5 in red for the same northern limit and yellow for total CryoSat-2 coverage. Overlapping period of the satellite missions is 11/2010–03/2012.



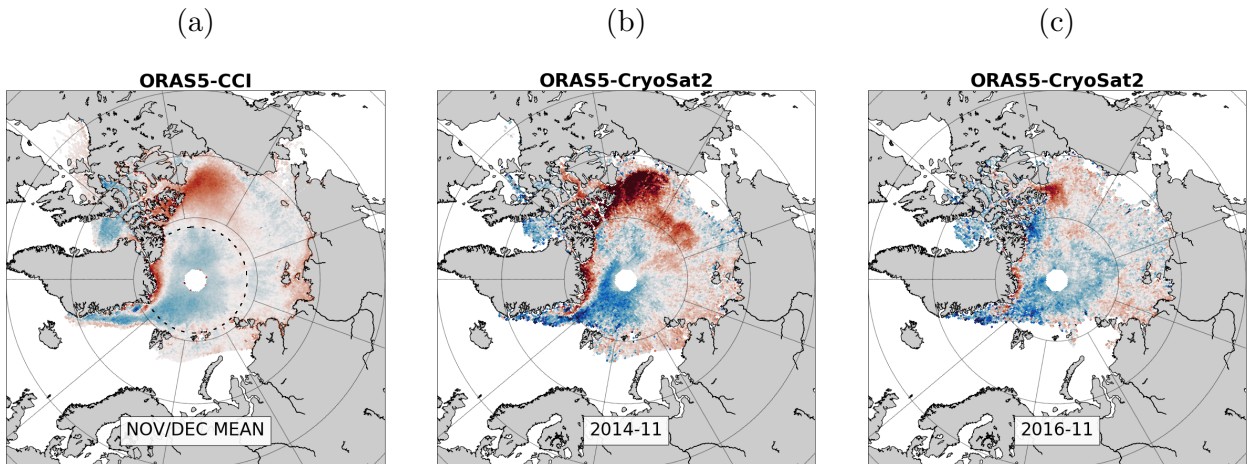

**Figure 2.** The average sea-ice thickness difference in autumn including ENVISAT and CryoSat-2 missions 2002–2017 south of 81.5 °
(dashed line), and CryoSat-2 2010–2017 north of 81.5 ° (a) the worst year (b) and the best year (c) of CryoSat-2 mission.

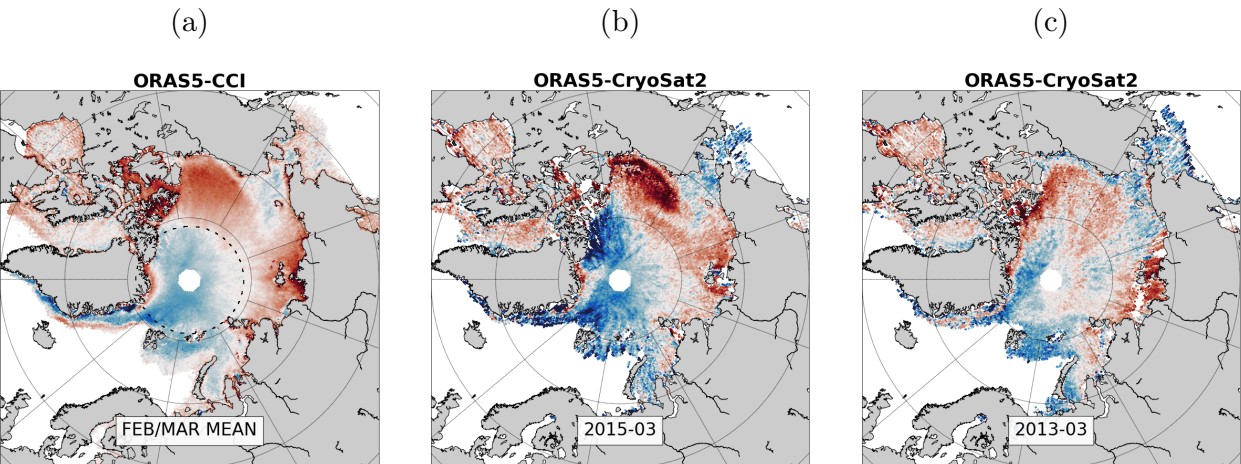

**Figure 3.** The average sea-ice thickness difference in spring including ENVISAT and CryoSat-2 missions 2002–2017 south of 81.5 ° (dashed line), and CryoSat-2 2010–2017 north of 81.5 ° (a) the worst year (b) and the best year (c) of CryoSat-2 mission.



**Figure 4.** Mean sea-ice thickness correlation between CCI and ORAS5 in Siberian side of the Arctic in two seasons. Mean is calculated for time period of total CCI dataset 2002–2017 for each month of the season separately. For the definitions of the regions see Table 2.



**Figure 5.** Mean sea-ice thickness correlation between CCI and ORAS5 in the North-American side of the Arctic in two seasons. Mean is calculated for time period of total CCI dataset 2002–2017 for each month of the season separately.





**Figure 6.** Time series of sea-ice volume (SIV) calculated over the grid cells with satellite data coverage between 45–81 °N for each month of the growth season from October to April. If the linear trend is not statistically significant, it is marked with dashed instead of continuous line. The trends in numbers are given in Table 3.





**Table 1.** Datasets used in this work and their time coverage. CCI consists of ENVISAT and CryoSat-2 datasets. From ORAS5 data the opa0 ensemble member is used. Between the starting month and ending month of CCI datasets, there are data for every month of the growing season (October–April).

| Data set | Time period |
|---|---|
| ENVISAT | 10/2002 – 03/2012 |
| CryoSat-2 | 11/2010 – 04/2017 |
| CCI | 10/2002 – 04/2017 |
| ORAS5 (opa0) | 01/2002 – 12/2017 |





**Table 2.** The sections of the Arctic used for the correlation comparison between ORAS5 and CCI datasets. Positive longitudes are east and negative west.

| Eastern Arctic | Longitudinal limits (°) | Western Arctic | Longitudinal limits (°) |
|---|---|---|---|
| Norwegian Sea | 0, 20 | Greenland Sea | 0, -40 |
| Barents Sea | 20, 60 | Baffin Bay | -40, -80 |
| Kara Sea | 60, 100 | Can. Arctic Archipelago | -80, -120 |
| Laptev Sea | 100, 140 | Beaufort Sea | -120, -160 |
| East Siberian Sea | 140, 180 | Chukchi Sea (& Bering Sea) | -160, -180 |



**Table 3.** Trends for sea-ice volume for each month of data coverage. ENVISAT period is 2002–2012, CryoSat-2 2010–2017, CCI and ORAS5 2002–2017. ORAS5 - CCI denotes the difference between ORAS5 and CCI trends. Units are in $10^3$ km$^3$ per decade. The trend is not statistically significant (p > 0.05) if marked with *.

| Data | Oct trend | Nov trend | Dec trend | Jan trend | Feb trend | Mar trend | Apr trend |
|------|-----------|-----------|-----------|-----------|-----------|-----------|-----------|
| ENVISAT | -1.83 | -1.33 | -1.41 | -1.21 | -1.13 | -1.30* | -0.80* |
| CryoSat-2 | 0.80* | 0.61* | 0.85* | 0.53* | 0.33* | -0.45* | -1.48* |
| CCI | -0.83 | -1.13 | -1.12 | -1.00 | -0.99 | -1.22 | -1.19 |
| ORAS5 | -1.14* | -0.74* | -0.53* | -0.68* | -0.24* | -0.47* | -0.70* |
| ORAS5 - CCI | -0.31 | 0.39 | 0.59 | 0.32 | 0.75 | 0.75 | 0.49 |