# Peer review of "Consistent variability but different spatial patterns between observed and reanalysed sea-ice thickness"

_The Cryosphere, 2019_

## Referee Comment (RC1) · Anonymous Referee #1 · 6 Feb 2020

The manuscript by Siponen and coauthors presents a validation of a recent coupled ice-ocean reanalysis product against a new sea ice thickness satellite climate record joining two altimeters, but only available during the cold season. The validation is rigorous, both the model and observations are at the state of the art, and the manuscript is well-written.

The exercise is unfortunately limited in scope - a monovariate validation - and does not reveal much more than was already known from previous intercomparison articles. The new products compared are certainly of higher quality, but the analysis does not benefit much from this.

[Figure]

This becomes cruelly evident when the authors state the objective of the paper: "Can the ESA CCI sea-ice thickness product be used for the validation of sea ice in the ORAS5 ocean reanalysis during the growth season?", which is not a scientific objective per se.

The results reveal contrasting findings, well summarized in the title, but the analysis is too shallow, only listing non-prioritized factors without pursuing any of them any more than was done in previous literature.

There are multiple ways the study could evolve into a more informative paper: pursue the ice drift issue by calculating volume fluxes between the different regions, extract the local ice production and melt from the model thermodynamics, the sea ice deformations and the data assimilation increments. The thermodynamics could be further pursued as well by comparing the model snow parameters to the Warren climatology and comparing the full yearly cycle of ice thickness against the Beaufort Gyre Exploration Project http://www.whoi.edu/beaufortgyre, these data are freely available. The validation of atmospheric and ocean parameters of relevance for sea ice could be included as well. There are many different directions this paper could evolve to become more informative.

Such additional analysis may represent significant work and I am not confident that this can be done during the review process, I therefore recommend rejection of the paper and resubmission when the analysis has better documented the likely causes for the differences between model and observations.

Detailed comments:

- Abstract l14: Is an RMSE of 1m a sign of good quality? What is the baseline for "good performance"?

- Abstract l15: the causes of differences should be prioritized.

- l97: when assimilating sea ice concentrations, what is the thickness of the "added

ice" and its snow depths?

- l102: What is the "central" ensemble member in this context?

- l126: The orbit of ENVISAT should replace the "track" of ENVISAT.

- l139: I understand that the model concentrations are disregarded here, so talking about "sea ice volume" is misleading because the model volume is a mix of model (thickness) and observations (concentrations). For the sake of clarity, the volume should be replaced by the average thickness.

- l179: Hudson Bay and other semi-enclosed areas receive no ice export from other places so it could be worth concentrating on these areas to evaluate the thermody-namics of the model separately from the dynamics.

- l185 / Table 2: The Norwegian Sea is by definition further south than Svalbard and ice-free. To avoid confusion, the sector should be renamed "Svalbard Area" or maybe "Norwegian Sector" referring to WMO MetOcean areas.

- l192: "coastal polynyas": If the model does not resolve these polynyas, what are the main processes for ice formation in the model?

- l232: The issue of interpolation to the observations or onto the model grid comes twice in the paper, is it important enough to deserve such attention, if yes please indicate the difference in number but if not this text can be shortened.

- l285: Level ice is subject to stresses and undergoes deformations like any other types of ice, so I don't see why there should be no leads in thin ice. The smaller freeboard could be an issue, though.

- l291: "could produce too large drift": you certainly have the model and observed drift available, so these should be compared directly rather than referring to some previous literature.

- l307: have you used negative snow depths in Hudson Bay? If not, what snow depth

have you used?

- l310: "which should happen soon". Please refer to ongoing research rather than wishful thinking. papers by Rostosky et al. 2018, or more recently Kilic et al. 2019 and Liu et al.:

Kilic, L., Tonboe, R. T., Prigent, C., and Heygster, G.: Estimating the snow depth, the snow–ice interface temperature, and the effective temperature of Arctic sea ice using Advanced Microwave Scanning Radiometer 2 and ice mass balance buoy data, The Cryosphere, 13, 1283–1296, https://doi.org/10.5194/tc-13-1283-2019, 2019.

Liu J. et al. 2019 Remote Sens. 2019, 11(23), 2864; https://doi.org/10.3390/rs11232864

- Section 5.3: As explained earlier, the comparison of mean thicknesses would make more sense when the model concentrations are ignored.

- l321: The anomalous events are well visible in the time series, no need to remove the trend.

- Figures 2 and 3 are missing the colour scale.

- Figure 6: the CryoSAT2 period is too short for any meaningful trend. These should be removed (also from Table 3)

- Table 3: Splitting the trends per month is not really informative as long as the seasonal cycle is not validated against in situ measurements.

Typos:

- l79: Taken -> taking

- l342: consisted -> consistent

---

## Referee Comment (RC2) · Anonymous Referee #2 · 9 Feb 2020

This paper presents a comparison between the relatively new ESA CCI sea ice thickness dataset (a combination of CryoSat-2 and ENVISAT freeboards), with sea ice thickness in the ECMWF's ORAS5 reanalysis. A simple RMSE/correlation analysis is used to compare the datasets, and a discussion of possible causes of discrepancies is included.

As stated at the end of the introduction: "The aim of the study is to give an answer to the question: Can the ESA CCI sea-ice thickness product be used for the validation of sea ice in the ORAS5 ocean reanalysis during the growth season? To answer this question, the mean sea-ice thickness as well as trends in sea-ice thickness and sea-ice

volume are compared, and their uncertainties are taken into account." Unfortunately, I think this aim is rather basic (the cited references doing similar things are often in the form of technical notes for their reanalyses), and more importantly, the analysis lacks the level of scientific robustness/completeness I would expect from a paper in The Cryosphere. The paper title is also pretty misleading. It's a comparison of a radar observed sea ice record and one ocean reanalysis, so not anything near as complete as the title suggests.

Some more specific comments:

You need to try and quantify the uncertainty. Most of the 'uncertainty analysis' was just discussion about biases/discrepancies which were often very subjective and arbitrary.

There was a lot of subjectivity in the introduction and discussion throughout the manuscript too (e.g. the first line!). It's important to base scientific papers in objectives as much as possible. Another example (there were many more) - The Blanchard-Wriggleworth (2018) study is just one of many new studies looking at snow on sea ice and does not provide evidence of its contribution to sea ice thickness uncertainty. I think someone reading this paper would come away with a misleading idea about the state of knowledge in our field.

The comments about it being better than ICESat were pretty odd – sure it might have better temporal sampling but that doesn't make it a better validation dataset (e.g. it could be less accurate!).

What validation/assessment has already been done with CCI? There was only limited comments about other CS-2 derived sea ice thickness products and an attempt to quantify the uncertainty from the choice of retracking and other input assumptions.

What's the central ensemble member and why again was this chosen?

What exactly did the recent Tietsche and Zuo studies do and what have we learnt from this.

How is sea ice used in the ORAS5 reanalysis? How important do you think sea ice thickness biases are? Is the aim of ORAS5 to provide a reanalysis of sea ice, or is this just the boundary condition for the bigger focus of providing an ocean reanalysis?

How does ORAS5 compare to other ocean/global reanalyses in terms of it's sea ice model/assimilation approach etc? This larger context would make the paper much more illuminating.

---

## Author Comment (AC1) · 7 Mar 2020

We thank you for your comments which we respond below. While revising the manuscript, we will consider all points you raised and try to incorporate them.

*1. The manuscript by Siponen and coauthors presents a validation of a recent coupled ice-ocean reanalysis product against a new sea ice thickness satellite climate record joining two altimeters, but only available during the cold season. The validation is rigorous, both the model and observations are at the state of the art, and the manuscript is well-written.*

We thank Referee #1 for these encouraging comments and address the critical ones next.

*2. The exercise is unfortunately limited in scope – a monovariate validation – and does not reveal much more than was already known from previous intercomparison articles. The new products compared are certainly of higher quality, but the analysis does not benefit much from this.*

In our manuscript we present results based on a sea-ice thickness intercomparison between two datasets: the CCI satellite climate record (CDR) and the ECMWF ORAS5 ocean reanalysis. We think that the scientific results based on this first and rigorous comparison justifies the publication in The Cryosphere as they are relevant, original, novel and timely:

- Both the CCI CDR and ORAS5 are new products that have not been compared before, which makes our results original and novel. CCI CDR is the first long observational time series of its kind and agrees well with ORAS5. Because of the novelty of CCI CDR, study on how it compares with modelled estimates of SIT are of interest to the readers of TC. We should point out that at the moment peer-reviewed publications presenting the CCI sea ice thickness CDR do not exist which increases the value of our manuscript.

- There is an urgent need to better understand the decadal-scale variability of the Arctic sea ice in a rapidly warming climate. This makes our topic timely.

- Sea-ice thickness, one of the essential climate variables, aggregates sea-ice evolution due to both thermodynamic and dynamic processes, therefore being an excellent variable for evaluations and intercomparisons. This makes our choice of single variable relevant.

- Moreover, as ORAS5 generates initial conditions for the ECMWF extended- and long-range predictions, its assessment in terms of sea-ice thickness is a sensible

choice to gain a good understanding of its skill in the Arctic. As both CCI and ORAS5 are expected to have many users in the scientific community, it is important to quickly disseminate information on their mutual uncertainties on a suitable forum, such as The Cryosphere. This makes our manuscript timely.

The main new findings of our study are:

- Both CCI and ORAS5 realistically capture Arctic-wide inter-annual variability and decadal trends of sea-ice thickness, so they are suitable for subsequent studies.

- CCI and ORAS5 also agree generally well, revealed by relatively small RMSE, although there are seasons and years when they disagree. This result provides a new focus for elaborative model performance studies.

We are keen to expand the scope of our paper by analysing an ensemble of seven ocean reanalysis which can be calculated from individual product data available at https://icdc.cen.uni-hamburg.de/1/daten/reanalysis-ocean/oraip.html. This will address the issue of reanalysis uncertainty and indicate how significantly reanalyses vary from observations. We will also explore the effect of snow on ice on sea-ice volume based on those ocean reanalyses with snow data available. This will address probably the most significant issue of uncertainty due to the large effect of snow thickness on sea-ice volume. We argue that the facts listed above and the additional analysis we propose will expand the paper scope so it is clearly appropriate for the publication in the Cryosphere.

*3. This becomes cruelly evident when the authors state the objective of the paper: "Can the ESA CCI sea-ice thickness product be used for the validation of sea ice in the ORAS5 ocean reanalysis during the growth season?", which is not a scientific objective per se.*

True, this is not a scientific objective, but a practical and useful one. We were understanding that it would also be a sufficient one for a research article in The Cryosphere. As this may not be the case, it is also possible to state a more scientific objective as "to identify regions, seasons and years where and when the efforts to simulate the observed Arctic sea-ice variability should focus on to improve the realism of modelled ice thickness." As the significance of physical mechanisms varies seasonally, our results should assist model developers to proceed towards this goal.

*4. The results reveal contrasting findings, well summarized in the title, but the analysis is too shallow, only listing non-prioritized factors without pursuing any of them any more than was done in previous literature.*

We understood that our main results have been novel, at least to the extent explained above. If Referee 1 disagrees, we would like to invite her/him to provide us with the relevant references, please. This would be very helpful.

*5. There are multiple ways the study could evolve into a more informative paper: pursue the ice drift issue by calculating volume fluxes between the different regions, extract the local ice production and melt from the model thermodynamics, the sea ice deformations and the data assimilation increments. The thermodynamics could be further pursued as well by comparing the model snow parameters to the Warren climatology and comparing the full yearly cycle of ice thickness against the Beaufort Gyre Exploration Project http://www.whoi.edu/beaufortgyre, these data are freely available. The validation of atmospheric and ocean parameters of relevance for sea ice could be included as well. There are many different directions this paper could evolve to become more informative.*

Thank you, these are good and interesting suggestions and we are happy to explore them. However, we need to keep the scope of the manuscript reasonably limited. As mentioned above, we will expand the scope of the paper to explore the uncertainty among the ocean reanalyses, and the satellite data. In terms of ocean reanalysis

ensemble, we will add results based on its mean and spread enabling statistical significance testing of reanalysis-observational differences.

*6. Such additional analysis may represent significant work and I am not confident that this can be done during the review process, I therefore recommend rejection of the paper and resubmission when the analysis has better documented the likely causes for the differences between model and observations.*

Fair enough. We need to assess the amount of extra work while completing the manuscript in a reasonable time to provide its main results to the scientific community in a timely manner. We also think that the extra analysis we suggest can be carried out during the review process and will better quantify data uncertainties and explain model-observation differences.

---

## Author Comment (AC2) · 7 Mar 2020

Thank you for your comments on this manuscript. Please find below our detailed responses to them. In the revised manuscript, we will take all the points you raised into account.

*1. This paper presents a comparison between the relatively new ESA CCI sea ice thickness dataset (a combination of CryoSat-2 and ENVISAT freeboards), with sea ice thickness in the ECMWF's ORAS5 reanalysis. A simple RMSE/correlation analysis is used to compare the datasets, and a discussion of possible causes of discrepancies is included.*

[Figure]

We thank Referee #2 for her/his comments and suggestions. We think that even if our statistical analysis is described as simple, this does not reduce its value.

*2. As stated at the end of the introduction: "The aim of the study is to give an answer to the question: Can the ESA CCI sea-ice thickness product be used for the validation of sea ice in the ORAS5 ocean reanalysis during the growth season? To answer this question, the mean sea-ice thickness as well as trends in sea-ice thickness and sea-ice volume are compared, and their uncertainties are taken into account." Unfortunately, I think this aim is rather basic (the cited references doing similar things are often in the form of technical notes for their reanalyses), and more importantly, the analysis lacks the level of scientific robustness/completeness I would expect from a paper in The Cryosphere. The paper title is also pretty misleading. It's a comparison of a radar observed sea ice record and one ocean reanalysis, so not anything near as complete as the title suggests.*

We agree, our aim is a basic one, but we also think it is an important one, as we explained in our response to the 2. comment by Referee #1. We would like to point out that we use new products (CCI and ORAS5) which are higher quality than the previous ones used in this context, and that our analysis methods are vigorous, as pointed out in the 1. comment by Referee #1. We also believe that our results have such a significance, as explained in our response to the 2. comment by Referee #1, that they should be published in The Cryosphere, rather than in a technical report, which would be missed by most scientists interested in the topic, we think.

*3. Some more specific comments:*

*You need to try and quantify the uncertainty. Most of the 'uncertainty analysis' was just discussion about biases/discrepancies which were often very subjective and arbitrary. There was a lot of subjectivity in the introduction and discussion throughout the manuscript too (e.g. the first line!). It's important to base scientific papers in objectives as much as possible. Another example (there were many more) - The Blanchard-*

*Wriggleworth (2018) study is just one of many new studies looking at snow on sea ice and does not provide evidence of its contribution to sea ice thickness uncertainty. I think someone reading this paper would come away with a misleading idea about the state of knowledge in our field.*

We are happy to follow this important suggestion, keeping in mind that our analysis is vigorous. Before beginning, we would like to invite Referee #2 to provide us with the relevant references in addition to Blanchard-Wriggleworth (2018), please. This would be very helpful.

In terms of the quantification of the uncertainty, we will expand the scope of our paper by analysing an ensemble of seven ocean reanalysis which can be calculated from individual product data available at https://icdc.cen.uni-hamburg.de/1/daten/reanalysis-ocean/oraip.html. This will address the issue of reanalysis uncertainty and indicate how significantly reanalyses vary from the observations. We will also explore the effect of snow on ice on sea-ice volume based on ocean reanalyses. This will address probably the most significant issue of uncertainty due to the large effect of snow thickness on sea-ice volume.

*The comments about it being better than ICESat were pretty odd – sure it might have better temporal sampling but that doesn't make it a better validation dataset (e.g. it could be less accurate!). What validation/assessment has already been done with CCI? There were only limited comments about other CS-2 derived sea ice thickness products and an attempt to quantify the uncertainty from the choice of re-tracking and other input assumptions.*

We and Referee #1 (2nd comment) think that the products we analysed, including CCI, are certainly higher quality than the previously used ones, such as ICESat-1. ICESat-1 was not good because of the temporal coverage. Specifically, we state on line 238 that "the CCI product is a better validation product in the sense of temporal coverage" and on line 338 that "It [CCI] is better than ICESat due to its temporal and spatial coverage

over the whole growing season". We do not actually say that CCI is a better validation dataset than ICESat. We will reword these sentences to clarify this and not to mislead the reader. We will also reformulate the title to be less misleading keeping in mind that Referee 1 describes that the title summarizes well the findings of the study.

*What's the central ensemble member and why again was this chosen?*

The ORAS5 central ensemble member was selected as the first ensemble member, marked as ora0, out of six. We carried out the analysis for this member, and as the sea ice is essentially similar between the members, we decided not to include other members in the analysis. In other words, in terms of results, it did not matter which member was chosen.

*What exactly did the recent Tietsche and Zuo studies do and what have we learnt from this?*

We will provide more details to better clarify what we've learnt from this.

*How is sea ice used in the ORAS5 reanalysis?*

In terms of sea-ice observations, ORAS5 assimilates sea-ice concentration from OS-TIA (see line 97), but no other sea-ice parameters. To clarify this, we will add some text to specify which sea-ice observations are used in ORAS5.

*How important do you think sea ice thickness biases are?*

This is a good question, we'd say they are important and in particular RMSE values in the order of several tens of centimetres or greater appear large compared to Arctic mean ice thicknesses. As sea-ice thickness biases result from disagreements in processes related to sea-ice evolution, we'd argue that they're manifestations of model and observational uncertainties and errors. These need to be first quantified and then addressed to improve the representation of processes related to sea-ice evolution in forecasting systems.

*Is the aim of ORAS5 to provide a reanalysis of sea ice, or is this just the boundary condition for the bigger focus of providing an ocean reanalysis?*

Sea ice is an integral and dynamic part of ORAS5, it is not just a boundary condition. However, given that ORAS5 is global, sea ice is certainly not the main focus of ORAS5.

*How does ORAS5 compare to other ocean/global reanalyses in terms of it's sea ice model/assimilation approach etc?*

The answer to this question can be found from Table 1 by Chevallier et al. (2017), which is cited in the manuscript. In general terms, compared to other state-of-the-art ocean reanalyses, ORAS5 has a rather simple sea-ice model (LIM2) but a mature assimilation of sea-ice concentration. Like almost all other reanalyses, sea-ice thickness is not assimilated, and therefore the covariance between sea-ice concentration and thickness are not properly accounted for.

*This larger context would make the paper much more illuminating.*

Thank you for these other important questions. We will address these in the revised manuscript.